# Experiences of food insecurity in the Roma population before and during the COVID-19 lockdown in Spain

Panmela Soares[1,2], Betlem Heras Molins[3], Mª Asunción Martínez Milán[1], Mª Félix Rodríguez Camacho[4], Vicente Clemente-Gómez[2], Iris Comino[2]*, Mª Carmen Davó-Blanes[1,2]

1 Departamento de Enfermería Comunitaria, Medicina Preventiva y Salud Pública e Historia de la Ciencia, Universidad de Alicante, Alicante, Comunidad Valenciana, España, 2 Grupo investigación Salud Pública, Universidad de Alicante, Alicante, Comunidad Valenciana, España, 3 Estudiante del Máster Universitario de Nutrición y Alimentación, Universidad de Alicante, Alicante, Comunidad Valenciana, España, 4 Federación Autónoma de Asociaciones Gitanas de la Comunidad Valenciana, Alicante, Comunidad Valenciana, España

* iriscomino@gmail.com

**Data Availability Statement:** All relevant data are within the manuscript and its Supporting information files.

## Abstract

### Objective

To explore the food insecurity experienced by the Roma population of the Valencian Community (Spain) and the effect of the COVID-19 lockdown.

### Method

Quantitative, cross-sectional exploratory study using a questionnaire that collected information on socioeconomic status and situations of food insecurity experienced before and during lockdown, based on the Food Insecurity Experience Scale of the United Nations Food and Agriculture Organization. The questionnaire was applied by health workers from the Roma community with people over 18 years of age. A descriptive analysis was carried out stratifying by sex, calculating Chi-square test to identify differences in the variables of the experiences of food insecurity.

### Results

468 people participated (57.1% women/42.9% men) who expressed: worry about a lack of food (67.3%); eating the same type of food (37.2%); not being able to eat healthy foods (34.4%); feeling hungry and not being able to eat (9.6%). Around 2.1 percent stated that they could not eat for a whole day, and 65 percent reported that they had to ask for or provide help to be able to eat. When stratifying by sex, it was found that women had more experiences of food insecurity. Except in the case of having stopped eating for a full day, where the percentage remained constant, an increase was observed in the other experiences of food insecurity during lockdown.

**Funding:** This work was supported by the Chair of Gipsy Culture, University of Alicante. Call: BOUA 8/10/2019. Text editing costs were supported by AICO, Generalitat Valenciana (2022- 2024). Call: CIAICO/2021/019. The funders had no role in study design, data collection and analysis, decision to publish, or preparation of the manuscript.

**Competing interests:** The authors have declared that no competing interests exist.

## Conclusions

A large part of the Roma population studied, especially women, experienced situations of food insecurity before COVID-19 that were aggravated during lockdown. This situation was compensated for by community support networks.

## Introduction

Food security has been recognized since 1996 by the United Nations as the right of all people to have permanent physical, economic and social access to quantitatively and qualitatively adequate food [1]. However, not all people can access a safe, nutritious diet that is consistent with their food preferences that allows them to lead a healthy life, and find themselves in a situation of food insecurity [2, 3]. It is estimated that, in 2021, around 30 percent of the world population (2.3 billion people) suffered from moderate or severe food insecurity. and 11.7 percent (923.7 million people) from severe food insecurity [4].

Poverty and inequality have been identified as structural causes of food insecurity [5, 6]. It is for this reason that studies on this subject are concentrated in low-income countries. However, food insecurity also affects high-income countries, specifically vulnerable populations [7, 8]. Furthermore, there is evidence that food insecurity affects women more than men. In 2021 an estimated 31.9% of women and 27.6% of men around the world experienced moderate or severe food insecurity [4]. This difference seems to be related to gender inequalities that affect women's access to food and other resources [9].

Gender is a strong determinant of most health outcomes [10, 11]. Patriarchy, characterised by the domination of women by men, restricts women's access to social privileges, labour market and economic resources, and assigns them unpaid domestic responsibilities. These are all social determinants that influence their health. These gender inequalities in power and in paid and unpaid work are the result of gender stereotypes and roles that begin in childhood and continue throughout life [12–14].

Given the magnitude of poverty and food insecurity in the world, and taking into account their relationship with gender inequalities, the goals of the UN 2030 Agenda for Sustainable Development propose to eradicate extreme poverty for all people in the world, create a world without hunger, and achieve gender equality and empower all women and girls [15].

The Roma population is a group that is in a situation of vulnerability. It is the largest ethnic minority in the European Union, and Spain is the second country in the region with the highest number of Roma people [16]. However, it is difficult to know the real geographical extension and distribution of the Roma population due to the lack of an ethnic minority census in Spain. For this reason, in many studies, it is necessary to rely on organizations that work with the Roma population in order to contact the population and obtain this information [17].

Social and economic inequalities have an impact on Roma health [18]. Recent data indicates that in Spain, the Roma population has suffered a significant setback in living conditions, with increasing levels of food insecurity as a result of the 2008 economic crisis [19]. The effects on health derived from an inadequate diet have been identified in the entire population, however, malnutrition, both due to deficit and excess of food, is found in families in situations of socioeconomic vulnerability [20].

The COVID-19 pandemic affected a large number of people worldwide. In September 2021, almost 223 million people were infected and more than 4 million died. In Spain, more than 5 million people were infected, and more than 87,000 people had died [21].

In the absence of effective treatment and in an attempt to curb the spread of the virus, governments in several countries began implementing mobility restriction measures proposed by the World Health Organisation in 2020. Closing schools, restaurants, shops and banning the movement of people were among the measures adopted. In Spain, the state of emergency related to the COVID-19 pandemic resulted in the lockdown of the population for three months. Movement restrictions affected economic activities, especially in the informal subsistence sector [22, 23].

The measures taken to curb the spread of the virus led to increased poverty and social inequalities, aggravating the situation of food insecurity in various regions of the world and in different populations [24]. However, there is little research on how the COVID-19 pandemic affected the food security of the Roma population.

Given the situation of vulnerability of the Roma population and the repercussion that the pandemic may have had on access to adequate food, the objective of this paper is to explore the food insecurity experienced by the Roma population in the Valencian Community (Spain), and the effect of the COVID-19 lockdown. The results of the study can inform the development of strategies to address food insecurity among vulnerable minorities, especially in times of crisis such as the COVID-19 pandemic.

## Methodology

A convenience sample of approximately 400 individuals over 18 years of age residing in the Valencian Community (Spain) was estimated.

We carried out an exploratory, cross-sectional study using a quantitative approach via a questionnaire designed for the Roma population. A convenience sample of approximately 400 individuals aged 18 and older residing in the Valencian Community (Spain) was estimated. A balance was sought in the number of participants according to sex and province of residence (Castellon, Valencia y Alicante). Contact with the participants took place through the Autonomous Federation of Roma Associations of the Valencian Community (FAGA). The selection process of the participants was carried out by means of telephone contact with the people attended by the collaborating associations who met the inclusion criteria.

For data collection, a questionnaire was prepared that was reviewed by health workers from the Roma community and piloted with 10 participants from the city of Alicante. It was adapted to the population and context. The final questionnaire included questions on: 1. Socioeconomic and demographic characteristics; 2. Food insecurity situations experienced by the participants, based on the FAO Food Insecurity Experience Scale [25]. This scale is made up of eight questions related to people's access to adequate food. Two other questions were added, related to the need to request or provide help for feeding the community. The inclusion of these questions was proposed during the questionnaire revision process that was carried out by members of the Roma community that act as health workers.

With the aim of knowing the effect of the lockdown by COVID19 on food security, three response options were provided: a) Yes, since before the lockdown; b) Yes, during the lockdown; or c) No. In addition, if the respondent answered in the affirmative (yes before or during lockdown), they were asked if it was due to economic reasons or other reasons. The variables included in the questionnaire are listed in Table 1.

Health workers from the Roma community, previously trained by members of the research team, were in charge of applying the questionnaire. In the training process, the concept of food security was discussed, the questionnaire was reviewed and doubts were clarified, and researchers agreed on the procedure for the collection of information. Data collection was

**Table 1. The variables included in the questionnaire.**

| Socioeconomic and demographic characteristics | |
|---|---|
| Province | Castellón/ Valencia/ Alicante |
| Belongs to the Roma community | Yes/ No |
| Country of birth | España |
| | Rumanía |
| Who do you live with? | With partner/family |
| | Alone |
| Number of people in the household | 1 to four people |
| | More than four people |
| Head of household | Me and my partner/ other family members |
| | Me |
| | My partner |
| | Don't know/Don't answer |
| Person responsible for chores at home | Me and my partner/ other family members |
| | Me |
| | My partner/ other family members |
| Education level | Complete secondary education or professional training |
| | Incomplete secondary education |
| | Complete primary education |
| | Incomplete primary education |
| | Does not read or write |
| | University studies |
| Employment situation | Unemployed |
| | Working |
| | Domestic chores |
| | Retired |
| Occupation | Manual labor |
| | Street vending |
| | Technician |
| | Administrative |
| | Don't answer |
| Receives pension or subsidy | Yes/ No |
| Which pension or subsidy do you receive | Unemployment |
| | Disability |
| | Social supports |
| | Widow or retired |

| Experiences of food insecurity of the participants | | | |
|---|---|---|---|
| Have you worried about not having enough food to eat? | Yes, since before the lockdown | If yes, why? | For economic reasons |
| Were you unable to consume healthy foods, such as fruit, vegetables and legumes? | | | |
| Have you had to eat the same type of food? | Yes, during the lockdown | | Others |
| Have you had to stop eating breakfast, lunch or dinner? | | | |
| Have you eaten less than you thought you should eat? | No | | |
| Have you run out of food at home? | | | |
| Have you felt hungry but were not able to eat? | | | |
| Have you stopped eating during a full day? | | | |
| Have you had to ask for help from someone in order to eat? | | | |
| Have you had to help someone else so they could eat? | | | |

carried out between May 20 and June 30 of 2020. Taking into account the health situation, contact with the participants and the application of the questionnaire was carried out by telephone.

Before applying the questionnaire, the participants were informed of the objective of the study, anonymity was guaranteed, and their free and informed verbal consent was requested, as established in the research ethics protocols. The study was approved by the Ethics Committee of the University of Alicante (UA-2020-03-03) and by representatives of the collaborating Roma associations. Additional information regarding the ethical, cultural, and scientific considerations specific to inclusivity in global research is included in the S1 File.

The data were recorded in electronic templates and later, a descriptive analysis was carried out using SPSS statistical software (IBM Corp., Armonk, NY, USA). The Chi-square test was used to explore differences in the experiences of food insecurity between men and women (SPSS software). In addition, odds ratios and 95% confidence intervals were calculated using Epi InfoTM software (CDC.gov., Atlanta, GA, USA).

## Results

Table 2 shows the socioeconomic and demographic characteristics of the study population. A total of 468 people participated in the study, 57.1 percent women and 42.9 percent men. The percentage of participating people was similar in the three provinces of the Valencian Community (Alicante 31.4%. Castellón 34.4%. Valencia 34.2%). An estimated 95.7 percent acknowledged being a member of the Roma community and 4.3 percent. a resident of it.

Almost all the participants (95.1%) stated that they lived with a partner or with relatives in households made up of between 1 and 4 people (74.8%). They stated that economic and household responsibilities tended to be shared with their partner or other family members (43.6% and 46.8% respectively). Of the 32.1 percent who stated that they were in charge of the household's economic responsibilities alone, more than 70 percent were men. Of the 30.0 percent who declared that they were in charge of the household chores alone, almost all were women (91.7%).

Just under a third of the people surveyed (31.8%) said they had completed secondary education or professional training, 19.2 percent had primary education, and almost five percent did not know how to read or write, among whom there was a greater percentage of women (78.3%).

Regarding the employment situation, almost 40 percent were unemployed at the time of the survey and 26.5 percent were engaged in domestic chores, in this case the majority were women (96.0%). Regarding occupation, of the 27.4 percent who worked at that time, more than 60 percent performed manual labor and close to 30 percent were engaged in street vending.

About 34.8 percent of the participants received some subsidy, the majority because of unemployment (39.3%), disability (26.4%) and social assistance (25.8%).

Table 3 shows the experiences of food insecurity experienced in the last year by the participants. About 67.3 percent stated that they felt worried about not having enough food to eat; more than a third had always had to eat the same type of food (37.2%), had not been able to eat healthy foods (34.4%), or had eaten less than they thought they should eat (31.0%). Less than 20 percent of the participants claimed to have run out of food at home (19.2%), to have missed breakfast, lunch or dinner (11.1%) and/or to have felt hungry and not been able to eat (9.6%). Regarding community support networks, 63.7 percent stated that they had to ask someone for help to be able to eat, and 67.7 percent had to provide it. In all cases the main reason was due to economic issues.

**Table 2. Socioeconomic and demographic characteristics of the study population.**

| Variables | | Total N (%) | Women n (%) | Men n (%) |
|---|---|---|---|---|
| | | **468 (100)** | **267 (57,1)** | **201 (42,9)** |
| Province | Castellón | 161 (34,4) | 111 (68,9) | 50 (31,1) |
| | Valencia | 160 (34,2) | 89 (55,6) | 71 (44,4) |
| | Alicante | 147 (31,4) | 67 (45,6) | 80 (54,4) |
| Belongs to the Roma community | Yes | 448 (95,7) | 251 (56,0) | 197 (44,0) |
| | No (neighbors) | 20 (4,3) | 16 (80,0) | 4 (20,0) |
| Country of birth | España | 449 (95,9) | 256 (57,0) | 193 (43,0) |
| | Rumanía | 19 (4,1) | 11 (57,9) | 8 (42,1) |
| Who do you live with? | With partner/family | 445 (95,1) | 253 (56,9) | 192 (43,1) |
| | Alone | 23 (4,9) | 14 (60,9) | 9 (39,1) |
| Number of people in the household | 1 to four people | 350 (74,8) | 207 (59,1) | 143 (40,9) |
| | More than four people | 118 (25,2) | 60 (50,8) | 58 (49,2) |
| Head of household | Me and my partner/ other family members | 204 (43,6) | 130 (63,7) | 74 (36,3) |
| | Me | 150 (32,1) | 35 (23,3) | 115 (76,7) |
| | My partner | 112 (23,9) | 100 (89,3) | 12 (10,7) |
| | Don't know/Don't answer | 2 (0,4) | 2 (100,0) | 0 (0,0) |
| Person responsible for chores at home | Me and my partner/ other family members | 219 (46,8) | 125 (57,0) | 94 (43,0) |
| | Me | 144 (30,8) | 132 (91,7) | 12 (8,3) |
| | My partner/ other family members | 105 (22,4) | 10 (9,5) | 95 (90,5) |
| Education level | Complete secondary education or professional training | 149 (31,8) | 70 (47,0) | 79 (53,0) |
| | Incomplete secondary education | 121 (25,9) | 70 (57,9) | 51 (42,1) |
| | Complete primary education | 90 (19,2) | 62 (68,9) | 28 (31,1) |
| | Incomplete primary education | 80 (17,1) | 42 (52,5) | 38 (47,5) |
| | Does not read or write | 23 (4,9) | 18 (78,3) | 5 (21,7) |
| | University studies | 5 (1,1) | 5 (100,0) | 0 (0,0) |
| Employment situation | Unemployed | 178 (38,1) | 74 (41,6) | 104 (58,4) |
| | Working | 128 (27,4) | 56 (43,7) | 72 (56,3) |
| | Domestic chores | 124 (26,5) | 119 (96,0) | 5 (4,0) |
| | Retired | 38 (8,1) | 18 (47,4) | 20 (52,6) |
| [a]Occupation (N = 128) | Manual labor | 74 (62,7) | 30 (40,5) | 44 (59,5) |
| | Street vending | 32 (27,1) | 10 (31,3) | 22 (68,8) |
| | Technician | 9 (7,6) | 7 (77,8) | 2 (22,2) |
| | Administrative | 3 (2,5) | 3 (100,0) | 0 (0,0) |
| | Don't answer | 10 (0,1) | 6 (60,0) | 4 (40,0) |
| Receives pension or subsidy | Yes | 163 (34,8) | 95 (58,3) | 68 (41,7) |
| | No | 305 (65,2) | 172 (56,4) | 133 (43,6) |
| [b]Which pension or subsidy do you receive (N = 163) | Unemployment | 64 (39,3) | 31 (48,4) | 33 (51,6) |
| | Disability | 43 (26,4) | 24 (55,8) | 19 (44,2) |
| | Social supports | 42 (25,8) | 31 (73,8) | 11 (26,2) |
| | Widow or retired | 14 (8,6) | 9 (64,3) | 5 (35,7) |

[a] Question answered by participants who were working at the time of the data collection.

[b] Question answered by participants who were receiving a pension or subsidy at the time of the interview.

When stratifying the results by sex, women experienced more food insecurity than men, with significant differences in the fact of having felt hungry and not being able to eat (73.3%), not having breakfast, lunch or eating (73. 1%) less than what they thought they should have eaten (64.8%) and not being able to eat healthy foods (64.0%) (p<0.05).

**Table 3. Experiences of food insecurity of the participants.**

| During the past years, at some time: | | Total N = 468 n (%) | Woman n = 267 n (%) | Man n = 201 n (%) | p | OR (CI) |
|---|---|---|---|---|---|---|
| Have you worried about not having enough food to eat? | Yes | 315 (67,3) | 181 (57,5) | 134 (42,5) | 0,798 | 1,05 (0,71–1,55) |
| | No | 153 (32,7) | 86 (56,2) | 67 (43,8) | | |
| If yes, why? | For economic reasons | 302 (95,9) | 177 (58,6) | 125 (41,4) | | - |
| | Others | 13 (4,1) | 4 (30,8) | 9 (69,2) | | |
| Were you unable to consume healthy foods, such as fruit, vegetables and legumes? | Yes | 161 (34,4) | 103 (64) | 58 (36) | 0,028* | 1,55 (1,05–2,29) |
| | No | 307 (65,6) | 164 (53,4) | 143 (46,6) | | |
| If yes, why? | For economic reasons | 153 (95,0) | 98 (64,0) | 55 (36,0) | | - |
| | Others | 8 (5,0) | 5 (62,5) | 3 (37,5) | | |
| Have you had to eat the same type of food? | Yes | 174 (37,2) | 108 (62,1) | 66 (37,9) | 0,092 | 1,39 (0,95–2,04) |
| | No | 294 (62,8) | 159 (54,1) | 135 (45,9) | | |
| If yes, why? | For economic reasons | 156 (89,7) | 99 (63,5) | 57 (36,5) | | - |
| | Others | 18 (10,3) | 9 (50,0) | 9 (50,0) | | |
| Have you had to stop eating breakfast, lunch or dinner? | Yes | 52 (11,1) | 38 (73,1) | 14 (26,9) | 0,013* | 2,22 (1,17–4,21) |
| | No | 416 (88,9) | 229 (55,0) | 187 (45,0) | | |
| If yes, why? | For economic reasons | 49 (94,2) | 36 (73,5) | 13 (26,5) | | - |
| | Others | 3 (5,8) | 2 (66,7) | 1 (33,3) | | |
| Have you eaten less than you thought you should eat? | Yes | 145 (31,0) | 94 (64,8) | 51 (35,2) | 0,023* | 1,6 (1,07–2,4) |
| | No | 323 (69,0) | 173 (53,6) | 150 (46,4) | | |
| If yes, why? | For economic reasons | 139 (95,9) | 89 (64,0) | 50 (36,0) | | - |
| | Others | 6 (4,1) | 5 (83,3) | 1 (16,7) | | |
| Have you run out of food at home? | Yes | 90 (19,2) | 54 (60,0) | 36 (40,0) | 0,529 | 1,16 (0,73–1,86) |
| | No | 378 (80,8) | 213 (56,3) | 165 (43,7) | | |
| If yes, why? | For economic reasons | 88 (97,8) | 53 (60,2) | 35 (39,8) | | - |
| | Others | 2 (2,2) | 1 (50,0) | 1 (50,0) | | |
| Have you felt hungry but were not able to eat? | Yes | 45 (9,6) | 33 (73,3) | 12 (26,7) | 0,020* | 2,22 (1,12–4,42) |
| | No | 423 (90,4) | 234 (55,3) | 189 (44,7) | | |
| If yes, why? | For economic reasons | 44 (97,8) | 33 (75,0) | 11 (25,0) | | - |
| | Others | 1 (2,2) | 0 (0) | 1 (100) | | |
| Have you stopped eating during a full day? | Yes | 10 (2,1) | 8 (80,0) | 2 (20,0) | 0,138 | 3,07 (0,65–14,63) |
| | No | 458 (97,9) | 259 (56,6) | 199 (43,4) | | |

(*Continued*)

**Table 3.** (Continued)

| During the past years, at some time: | | Total | Woman | Man | p | OR (CI) |
|---|---|---|---|---|---|---|
| | | N = 468 | n = 267 | n = 201 | | |
| | | n (%) | n (%) | n (%) | | |
| If yes, why? | For economic reasons | 8 (80,0) | 6 (75,0) | 2 (25,0) | | - |
| | Others | 2 (20,0) | 2 (100) | 0 (0) | | |
| Have you had to ask for help from someone in order to eat? | Yes | 298 (63,7) | 172 (57,7) | 126 (42,3) | 0,700 | 1,08 (0,74–1,58) |
| | No | 170 (36,3) | 95 (55,9) | 75 (44,1) | | |
| If yes, why? | For economic reasons | 298 (100) | 172 (54,3) | 126 (45,7) | | - |
| | Others | 0 (0) | 0 (0) | 0 (0) | | |
| Have you had to help someone else so they could eat? | Yes | 317 (67,7) | 172 (54,3) | 145 (45,7) | 0,077 | 0,7 (0,47–1,04) |
| | No | 151 (32,3) | 95 (62,9) | 56 (37,1) | | |
| If yes, why? | For economic reasons | 314 (99,1) | 171 (54,5) | 143 (45,5) | | - |
| | Others | 3 (0,9) | 1 (33,3) | 2 (66,7) | | |

* p<0,05

Fig 1 shows the increase in the percentage of people participating in the study who experienced of food insecurity before and during lockdown. Except in the case of having stopped eating for a full day, where the percentage remained at 2.1 percent, an increase was observed in the other experiences of food insecurity. A greater number of participants stated that during the lockdown they had to eat the same type of food (increased from 21.4% to 37.2%); could not

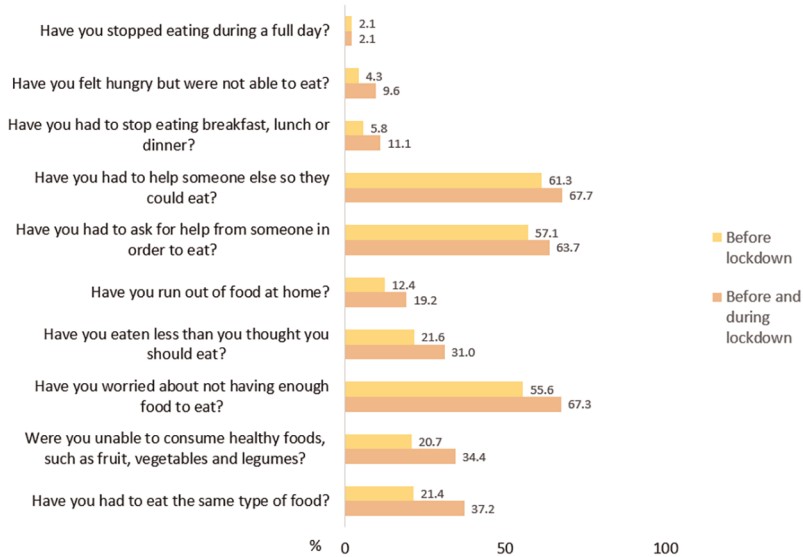

**Fig 1. Increase in the percentage of people participating in the study who experienced of food insecurity before and during lockdown (n = 468).**

eat healthy foods (from 20.7% to 34.4%); worried about not having enough food to eat (from 55.6% to 67.3%); and ate less than they thought they should eat (21.6% to 31.0%).

## Discussion

This study explored the food insecurity experienced by the Roma population of the Valencian Community and the effect of the COVID-19 lockdown. A significant part of the participants had some experience of food insecurity before COVID-19. This situation seems to be related to the context of vulnerability in which this population finds itself, which is manifest in their educational levels and employment situations. Food insecurity affects women more than men and was exacerbated during the lockdown period. However, support networks in the community seem to have helped to avoid more serious situations of food insecurity.

According to these results, a portion of the Roma population experienced some sort of food insecurity since before COVID-19. This is shown by their worry about not having enough food or not being able to consume varied and healthy foods. The context of vulnerability in which this population finds itself explains these results. A study carried out in Latin America revealed that low educational levels and limited social capital are associated with a greater probability of experiencing food insecurity [26]. In fact, the population with a low socioeconomic level has a less healthy diet [27]. The discrimination that the Roma population suffers in the labor market [28] influences their socioeconomic situation and, consequently, their eating patterns. In Spain, the Roma population has a harder time maintaining a healthy diet than the general population [22]. This situation can have a negative effect on health, given the relationship between an unhealthy eating diet and the development of chronic, non-communicable diseases [29].

When stratifying our results by sex, we observed that experiences of food insecurity seem to affect women more than men. The feeling of hunger, the difficulty in accessing food, missing a primary meal or eating less than one thinks they should have been prominent trends in the women in our study since before the lockdown. The report "The State of Food Security and Nutrition in the World 2020" revealed that the prevalence of food insecurity increased more among women [30]. The gender gap places women in a situation of social and economic disadvantage that affects their diet [31].

Our results suggest that the food insecurity experienced by the Roma population was aggravated during the lockdown period. These results are consistent with other studies carried out in Spain that show that the health crisis caused by COVID-19 has exacerbated situations of inequality. Part of the population experienced a reduction in economic income during the lockdown, which made it difficult to cover their basic needs. A report on the impact of COVID-19 on the Roma population in Spain during lockdown revealed that more than two-thirds of households had trouble meeting their food needs [22]. The measures imposed during the state of alarm, including the closure of non-essential activity, directly affected the commerce, business and service sectors in which a significant number of Roma families carry out their work activity. This situation further limited access to adequate food, which could consequently lead to a worsening in the quality of their diet and, therefore, their health [32].

However, as the literature also shows [22, 33], support networks in the community seem to help avoid more serious situations of food insecurity. In fact, most of the people participating in our study stated that at some point they had to have help from other people to be able to eat. It is important to take into account that community help and support for all Roma families in need is one of the values that characterizes the Roma ethnicity. There is evidence that this support has contributed to their survival during difficult times [34]. Given that social support

networks are a protective factor for food security [35], strengthening these networks should be a priority strategy to address food insecurity.

Measures to reduce the spread of the virus aggravated the food insecurity situation of the Roma population, highlighting the need for increased multisectoral action during health crises. In particular, gender mainstreaming in Roma-specific social protection actions could have minimised the effects of confinement on food security. A mapping of the population at social risk, as well as of the available community food safety nets, could facilitate the development of specific emergency food measures in times of crisis to ensure the food security of this population.

Taking into account that the people surveyed may have felt uncomfortable recognizing situations of food deprivation, it is possible that the results of this study have underestimated situations of food insecurity. To overcome this limitation, we used health workers from the community itself, because they are knowledgeable about the population. When interpreting the results of the study, it should be taken into account that they cannot be generalized to the Spanish Roma population as a whole, since the data come from a convenience sample. However, these results have made it possible to identify situations of food insecurity experienced by the vulnerable Roma population in the Valencian community.

In conclusion, a large part of the Roma population studied, especially women, experienced situations of food insecurity before COVID-19 that were aggravated during lockdown. This situation was compensated for by their community support networks.

## Supporting information

**S1 File. Inclusivity in global research questionnaire.**
(DOCX)

**S2 File. The values used to build graphs.**
(XLSX)

## Acknowledgments

To Autonomous Federation of Roma Associations of the Valencian Community (FAGA) and its health agents for their support in data collection.

## Author Contributions

**Conceptualization:** Panmela Soares, Mª Carmen Davó-Blanes.

**Data curation:** Vicente Clemente-Gómez.

**Formal analysis:** Panmela Soares, Betlem Heras Molins, Vicente Clemente-Gómez.

**Funding acquisition:** Mª Carmen Davó-Blanes.

**Investigation:** Mª Félix Rodríguez Camacho, Iris Comino.

**Methodology:** Panmela Soares, Mª Félix Rodríguez Camacho, Mª Carmen Davó-Blanes.

**Project administration:** Mª Carmen Davó-Blanes.

**Supervision:** Mª Carmen Davó-Blanes.

**Visualization:** Vicente Clemente-Gómez.

**Writing – original draft:** Panmela Soares, Betlem Heras Molins.

**Writing – review & editing:** Mª Asunción Martínez Milán, Mª Félix Rodríguez Camacho, Vicente Clemente-Gómez, Iris Comino, Mª Carmen Davó-Blanes.

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
