## [Decision Letter · Decision Letter 0]

30 Apr 2024

PONE-D-24-07420Experiences of Food Insecurity in the Roma Population Before and During the COVID-19 lockdown in SpainPLOS ONE

Dear Dr. Soares,

Thank you for submitting your manuscript to PLOS ONE. After careful consideration, we feel that it has merit but does not fully meet PLOS ONE’s publication criteria as it currently stands. Therefore, we invite you to submit a revised version of the manuscript that addresses the points raised during the review process.

We look forward to receiving your revised manuscript.

Kind regards,

George N Chidimbah Munthali

Academic Editor

PLOS ONE

Journal Requirements:

"This work was supported by the Chair of Gipsy Culture, University of Alicante. Call: BOUA 8/10/2019.

Text editing costs were supported by AICO, Generalitat Valenciana (2022- 2024). Call: CIAICO/2021/019."

5. Please provide a complete Data Availability Statement in the submission form, ensuring you include all necessary access information or a reason for why you are unable to make your data freely accessible. If your research concerns only data provided within your submission, please write "All data are in the manuscript and/or supporting information files" as your Data Availability Statement.

Additional Editor Comments:

Dear Authors

Kindly answer all the comments by the reviewers and resubmit the work.

Regards

Reviewers' comments:

Reviewer's Responses to Questions

**Comments to the Author**

1. Is the manuscript technically sound, and do the data support the conclusions?

Reviewer #1: Yes

Reviewer #2: Yes

2. Has the statistical analysis been performed appropriately and rigorously? 

Reviewer #1: Yes

Reviewer #2: Yes

3. Have the authors made all data underlying the findings in their manuscript fully available?

Reviewer #1: No

Reviewer #2: Yes

4. Is the manuscript presented in an intelligible fashion and written in standard English?

Reviewer #1: Yes

Reviewer #2: Yes

5. Review Comments to the Author

Reviewer #1: The study on Experiences of Food Insecurity in the Roma Population Before and During the COVID-19 lockdown in Spain si well elaborated and written, however the following comments should be considered.

1. Introduction

Kindly indicate the implication of the study after the objectives.

2. Methodology

How did the authors estimate the number of participants they used for the study? There is a need to provide the selection procedure and the estimation used. Again, the questionnaire should be provided as supplementary.

The abstract indicated a different number of participants are stated in the methodology. Please make the necessary corrections

Tables and figures cited in methodology: This table and figures seem to the results why citated in Materials and method. Please put the appropriate table for the methodology.

3. Results

Figure 2 seems not clear. Authors can present a different figure to make it presentable.

Reviewer #2: Though the study is restricted to a specific demographic population in Spain [that of the Roma ethnic group], the focus on the food insecurity impact of the Covid-19 pandemic and implications other such public health conditions could have in future resonates with experiences of a wide spectrum of populations globally, more so in low and middle income contexts. The study therefore contributes to the body of knowledge and literature on how to prepare for and handle food insecurity vulnerabilities as well as other impact they have like adverse health outcomes especially for minorities, and marginalized people across the globe. However some minor revisions are necessary before the paper can be published as follows:

Background

1. Authors need to place the paper within the scope of Sustainable development goal to give the paper a global appeal, specifically Goal # 2 on ending hunger and achieving food security and Goal # 3 on sustainable health for all.

2. In their background authors mention gender dis-aggregation with respect to food-insecurity within the Roma population with women considered disproportionately vulnerable on aggregate compared to their male counterparts. I feel however that authors need to go further to highlight potential correlates perpetrating the dichotomy and gender gap within the background.

3. A brief synopsis of the Covid-19 pandemic is necessary in the background to provide a snapshot of why it had to have the impact per-se on food insecurity. Perhaps its etiology, a bit of the incidence and prevalence, if not specifically in Europe then in Spain and even globally. Some highlight of public health measures that were used to avert the spread of the pandemic and the overall impact they had not only on food security but economies across the world would also suffice.

Methodology

The methodology is well articulated

Discussion

In their discussion, authors need to highlight what the Spanish government could have done to contain the food insecurity situation or crisis within this vulnerable minority and perhaps marginalized Roma ethnic group. From the findings, it seems they are a socially marginalized and vulnerable community and in essence deliberate government measures or interventions were necessary to ameliorate their plight. That could be included as part of study recommendations and preparedness for futture public health crisis for not only for the Roma populace obu other vulnerable groups in Europe.

6. PLOS authors have the option to publish the peer review history of their article (what does this mean?). If published, this will include your full peer review and any attached files.

Reviewer #1: No

Reviewer #2: **Yes: **Dr Marisen Mwale

---

## [Author Response · Author response to Decision Letter 0]

10 Jun 2024

Journal Requirements:

Response: The manuscript has been reviewed and conforms to the style requirements of PLOS ONE.

Response: We have completed and sent the questionnaire as supplementary information.

Response: The information has been reviewed. 

"This work was supported by the Chair of Gipsy Culture, University of Alicante. Call: BOUA 8/10/2019.

Text editing costs were supported by AICO, Generalitat Valenciana (2022- 2024). Call: CIAICO/2021/019." Please state what role the funders took in the study. If the funders had no role, please state: "The funders had no role in study design, data collection and analysis, decision to publish, or preparation of the manuscript." If this statement is not correct you must amend it as needed. Please include this amended Role of Funder statement in your cover letter; we will change the online submission form on your behalf.

Response: We have included information on the role of the funder in the cover letter of the manuscript. And also, in the funder's section. 

5. Please provide a complete Data Availability Statement in the submission form, ensuring you include all necessary access information or a reason for why you are unable to make your data freely accessible. If your research concerns only data provided within your submission, please write "All data are in the manuscript and/or supporting information files" as your Data Availability Statement.

Response: All data are in the manuscript and/or supporting information files. This statement has been included in the submission.

Response: We have removed this information from the statement’s section.

Response: The list of references has been revised.

Review Comments to the Author

Reviewer #1: The study on Experiences of Food Insecurity in the Roma Population Before and During the COVID-19 lockdown in Spain si well elaborated and written, however the following comments should be considered.

Response: We appreciated your time in reviewing this work. In this new version, we have incorporated the suggestions of the reviewers, and we believe that this has contributed to improvements in the manuscript.

1. Introduction

Kindly indicate the implication of the study after the objectives.

Response: We have included the implications of the study after the objective. See last paragraph of the introduction.

2. Methodology

How did the authors estimate the number of participants they used for the study? 

Response: A convenience sample was estimated, determined by budgetary considerations.

There is a need to provide the selection procedure and the estimation used. 

Response: In this new version, we have included in the methodology information on the selection procedure for participants. See second paragraph of the methodology. 

Again, the questionnaire should be provided as supplementary.

Response: We have included a new table in the methodology with the questionnaire variables. Please see table 1.

The abstract indicated a different number of participants are stated in the methodology. Please make the necessary corrections.

Response: A convenience sample of approximately 400 persons was estimated (as indicated in the methodology). In the end, a sample of 468 people was reached (as indicated in the summary and results). We have changed the wording of the methodology section to clarify this aspect. See the second paragraph of the methodology.

Tables and figures cited in methodology: This table and figures seem to the results why citated in Materials and method. Please put the appropriate table for the methodology. 

Response: In response to the reviewer's comments, we have included a new table in the methodology with the questionnaire variables. Please see table 1. 

3. Results

Figure 2 seems not clear. Authors can present a different figure to make it presentable.

Response: In this new version we have changed figure 1. See the figure 1.

Reviewer #2: Though the study is restricted to a specific demographic population in Spain [that of the Roma ethnic group], the focus on the food insecurity impact of the Covid-19 pandemic and implications other such public health conditions could have in future resonates with experiences of a wide spectrum of populations globally, more so in low and middle income contexts. The study therefore contributes to the body of knowledge and literature on how to prepare for and handle food insecurity vulnerabilities as well as other impact they have like adverse health outcomes especially for minorities, and marginalized people across the globe. However some minor revisions are necessary before the paper can be published as follows:

Response: We appreciated your time in reviewing this work. In this new version, we have incorporated the suggestions of the reviewers, and we believe that this has contributed to improvements in the manuscript.

Background

1. Authors need to place the paper within the scope of Sustainable development goal to give the paper a global appeal, specifically Goal # 2 on ending hunger and achieving food security and Goal # 3 on sustainable health for all.

Response: In response to the reviewer's comments, we provided information in the introduction related to the Sustainable development goal. Please see fourth paragraph of the introduction. 

2. In their background authors mention gender dis-aggregation with respect to food-insecurity within the Roma population with women considered disproportionately vulnerable on aggregate compared to their male counterparts. I feel however that authors need to go further to highlight potential correlates perpetrating the dichotomy and gender gap within the background.

Response: We have included in the introduction information on the factors related to the gender gap. See the third paragraph of the introduction. 

3. A brief synopsis of the Covid-19 pandemic is necessary in the background to provide a snapshot of why it had to have the impact per-se on food insecurity. Perhaps its etiology, a bit of the incidence and prevalence, if not specifically in Europe then in Spain and even globally. Some highlight of public health measures that were used to avert the spread of the pandemic and the overall impact they had not only on food security but economies across the world would also suffice.

Response: In response to the reviewer's comments, we provide information in the introduction on the Covid-19. In addition, we include information on public health measures taken to contain the spread of the virus. Please see the seventh, eighth and ninth paragraphs of the introduction. 

Discussion

In their discussion, authors need to highlight what the Spanish government could have done to contain the food insecurity situation or crisis within this vulnerable minority and perhaps marginalized Roma ethnic group. From the findings, it seems they are a socially marginalized and vulnerable community and in essence deliberate government measures or interventions were necessary to ameliorate their plight. That could be included as part of study recommendations and preparedness for futture public health crisis for not only for the Roma populace obu other vulnerable groups in Europe.

Response: We have included a paragraph with recommendations in the discussion. See sixth paragraph of the discussion.

---

## [Editor Report · Decision Letter 1]

18 Jun 2024

Experiences of Food Insecurity in the Roma Population Before and During the COVID19 lockdown in Spain

PONE-D-24-07420R1

Dear Dr. Iris Comino

We’re pleased to inform you that your manuscript has been judged scientifically suitable for publication and will be formally accepted for publication once it meets all outstanding technical requirements.

Kind regards,

George N Chidimbah Munthali

Academic Editor

PLOS ONE